# Pathogenicity Analysis of a Novel Variant in *GTPBP3* Causing Mitochondrial Disease and Systematic Literature Review

**DOI:** 10.3390/genes14030552

**Published:** 2023-02-22

**Authors:** Qin Zhang, Qianqian Ouyang, Jingjing Xiang, Hong Li, Haitao Lv, Yu An

**Affiliations:** 1Department of Cardiology, Children’s Hospital of Soochow University, Suzhou 215127, China; 2Center for Reproduction and Genetics, The Affiliated Suzhou Hospital of Nanjing Medical University, Suzhou 215008, China; 3Center for Reproduction and Genetics, Suzhou Municipal Hospital, Suzhou 215008, China; 4Human Phenome Institute, Zhangjiang Fudan International Innovation Center, MOE Key Laboratory of Contemporary Anthropology, Fudan University, Shanghai 201203, China

**Keywords:** *GTPBP3*, COXPD23, variants, taurine modification, mitochondrial diseases

## Abstract

Defect of *GTPBP3*, the human mitochondrial tRNA-modifying enzyme, can lead to Combined Oxidative Phosphorylation Deficiency 23 (COXPD23). Up to now, about 20 different variants of the *GTPBP3* gene have been reported; however, genotype–phenotype analysis has rarely been described. Here, we reported a 9-year-old boy with COXPD23 who presented with hyperlactatemia, hypertrophic cardiomyopathy, seizures, feeding difficulties, intellectual disability and motor developmental delay, and abnormal visual development. Biallelic pathogenic variants of the *GTPBP3* gene were identified in this boy, one novel variant c.1102dupC (p. Arg368Profs*22) inherited from the mother and the other known variant c.689A>C (p. Gln230Pro) inherited from father. We curated 18 COXPD23 patients with *GTPBP3* variants to investigate the genotype–phenotype correlation. We found that hyperlactatemia and cardiomyopathy were critical clinical features in COXPD23 and the average onset age was 1.7 years (3 months of age for the homozygote). Clinical classification of COXPD23 for the two types, severe and mild, was well described in this study. We observed arrhythmia and congestive heart failure frequently in the severe type with early childhood mortality, while developmental delay was mainly observed in the mild type. The proportion of homozygous variants (71.4%) significantly differed from that of compound heterozygous variants (18.1%) in the severe type. Compared with the variants in gnomAD, the proportion of LOFVs in *GTPBP3* was higher in COXPD23 patients (48.6% versus 8.9%, *p* < 0.0001 ****), and 31% of them were frameshift variants, showing the LOF mechanism of *GTPBP3*. Additionally, the variants in patients were significantly enriched in the TrmE-type G domain, indicating that the G domain was crucial for *GTPBP3* protein function. The TrmE-type G domain contained several significant motifs involved in the binding of guanine nucleotides and Mg^2+^, the hydrolysis of GTP, and the regulation of the functional status of GTPases. In conclusion, we reported a mild COXPD23 case with typical *GTPBP3*-related symptoms, including seizures and abnormal visual development seldom observed previously. Our study provides novel insight into understanding the clinical diagnosis and genetic counseling of patients with COXPD23 by exploring the genetic pathogenesis and genotype–phenotype correlation of COXPD23.

## 1. Introduction

Dysfunction of the mitochondrial respiratory chain and oxidative phosphorylation (OXPHOS) is responsible for developing mitochondrial disease with diverse clinical symptoms and heterogeneous genetic profiles. It is a rare clinical condition in children with an incidence rate of 5 to 15 cases per 100,000 people [1]. Typically, it dramatically impacts the brain, heart, liver, skeletal muscle, and other organs with high energy needs. One of the critical processes affecting mitochondrial function is mitochondrial protein synthesis. The mitochondrial DNA encodes all RNA elements of the mitochondrial translation system. In contrast, the nuclear DNA encodes all protein components (including aminoacyl-tRNA synthetases and tRNA-modified enzymes) [2]. Thus, both the nuclear and mitochondrial genomes control mitochondrial functioning.

In general, mature mitochondrial tRNA (mt-tRNA) must undergo extensive post-transcriptional modification, which is vital for the accurate and efficient translation of proteins in mitochondria [3]. Taurine modifications (τm5s2U and τm5U) exist in the wobble U34 position of the anticodon of five human mt-tRNAs (τm5U: mt-tRNA^Leu^ (UUR) and mt-tRNA^Trp^; τm5s2U: mt-tRNA^Lys^, mt-tRNA^Gln^, and mt-tRNA^Glu^), which is one of the few types of modifications presenting only in mt-tRNA [2]. 

The human *GTPBP3* gene is a nuclear coding gene located on chromosome 19p13.11, and it codes for the GTPBP3 protein, which is a highly conserved protein [4]. GTPBP3, dimerized through the N-terminal domain, interacts with MTO1 through the central helix domain to form a heterologous tetrameric complex in human cells [5]. The ꞵ-carbon of Ser, mediated by serine hydroxymethyltransferase 2 (SHMT2), or the α-carbon of Gly, mediated by the glycine cleavage system pathway, transfer to tetrahydrofolate (THF), forming 5,10-CH_2_-THF, which might be the one-carbon-unit donor in the first step of the τm5U modification reaction [6,7]. The GTPBP3-MTO1 complex employs taurine and 5,10-CH_2_-THF as substrates to mediate the taurine modifications of mt-tRNA in the presence of co-factors such as GTP, K^+^, and FAD [7]. However, the specific role of these co-factors in the response is currently unclear.

Defects in the GTPBP3 protein inhibit the taurine modification of mt-tRNA, leading to damage to mitochondrial translation, resulting in Combined Oxidative Phosphorylation Deficiency 23 (OMIM #608536, COXPD23). COXPD23 is an autosomal recessive disorder, usually occurring in infancy, which is characterized by the following major features: respiratory chain complex deficiency syndrome, lactic acidosis, hypertrophic cardiomyopathy, and encephalopathy [3].

Seventeen patients with COXPD23 have been reported worldwide, all of which are homozygous variants or compound heterozygous variants carrying 20 different variants of the *GTPBP3* gene. Here, we describe a case of COXPD23 driven by variants in *GTPBP3*, summarize his phenotypic characteristics with previously reported patients with pathogenic *GTPBP3* variants, and analyze and discuss the genetic pathogenesis and genotype–phenotype correlations of this disorder.

## 2. Results

### 2.1. Clinical Characteristics of the Patient

The boy was the first child of healthy parents and was delivered by cesarean section with a birth weight of 2.9 kg without asphyxia and dyspnea. The initial symptom was feeding difficulties on the third day after birth, which improved after hospitalization. Afterward, he was observed with motor, intellectual, and language developmental delays compared with his peers. After 16 months of life, the child could walk and run independently, but unstably, and could not bounce. We noticed no language development at the age of 3 years. Until he was 9 years old, he could speak 5–6 words with no clear pronunciation, understand only simple instructions, and play with his peers. There was no stereotyped behavior or developmental regression. Additionally, the proband had poor physical strength and was easily fatigued with poor vision and normal hearing.

He had experienced seizures since he was 8 months old and had taken anti-epileptic medication (Eitracetam) for 6 years. At 6 years old, the EEG revealed that the high-amplitude 2–4 hz sharp waves and slow waves distributed primarily in the right occipital portion were sporadic, occasionally affecting the bilateral frontal and occipital parts (Appendix A). No seizure episodes have happened since he was 7 years old.

Brain MRI demonstrated the emergence of small pieces of long T1, long T2, and high-signal flair shadows in the bilateral thalamus (Figure 1). Long shadows of the T2 signal could be observed in the right maxillary and the left sphenoid sinus. In conclusion, the patient had bilateral symmetrical damage to the thalamus and bilateral sinusitis.

Echocardiography at 8.5 years showed a significant thickening of the left ventricular wall with a thickness of 14.7 mm, compared with 5.5 mm 5 years earlier. Liver ultrasound showed slight enlargement of the liver, which was the same as that at 3 years of age.

At 10 years of age, he was symmetrical and short, measuring 119 cm (−2 sd to −3 sd) in height, weighing 22.5 kg, with windy ears, well-groomed dentition, poor occlusion and dislocation of the upper and lower teeth, and covering the lower teeth. Auscultation of the heart, lung, and abdomen revealed no apparent abnormalities.

Biochemical examination revealed hyperlactatemia (3–11.6 mmol/L, normal: 0–2.2 mmol/L), and normal hepatic and renal function, creatine kinase levels, and blood ammonia. The levels of amino acids and various acylcarnitines were normal by blood tandem–mass spectrometry. However, urine gas chromatography–mass spectrometry showed that the levels of lactic acid, 3-hydroxybutyric acid, 3-methylpentenoic acid, 3-hydroxyglutaric acid, and hippuric acid increased (Appendix A), suggesting mitochondrial energy metabolism disorders, ketosis, and nutritional disorders.

### 2.2. Molecular Genetic Analysis

Whole exome sequencing (WES) revealed that the patient carried compound heterozygous variants of *GTPBP3* (NM_032620.4). A novel variant c.1102dupC (p. Arg368Profs*22) in exon 7 of *GTPBP3* inherited from the maternal allele (Figure 2A) was predicted to induceNMD (nonsense-mediated decay), which was absent in gnomAD or other general populations, as a pathogenic variant (PVS1 (null variant), PM2_P) based on the criteria of ACMG guidelines. Another paternal variant c.689A>C (p.Gln230Pro) in exon 5 had a low minor allele frequency (0.0001) in east Asian in gnomAD (gnomADv2.1.1, https://gnomad.broadinstitute.org/) and alt. allele frequency (0.00027) in Chinese Han (PGG2.0, https://www.biosino.org/pgghan2/index) [8]. Three cross-species conservative analyses revealed evolutionary boundary constraints (GERP++:3.64, phastcons:1; phylop:3.219), but the Grantham distance was 76, changing Gln into Pro, which was considered more deleterious. In silico prediction analysis was mostly tolerable (REVEL 0.075, CADD 11.830) as BP4. Glutamine in the 230th position of *GTPBP3* (Q969Y2) was predicted in the MnmE helical domain (InterPro) and also located in the TrmE-type G domain of *GTPBP3*, which was a critical domain for *GTPBP3*. Four patients with the variant c.689A>C (p.Gln230Pro) were reported, and another allelic variant in trans was likely pathogenic. As a result, the criteria of PM3 based on ACMG guidelines can be upgraded to the PM3_strong level. Therefore, the variant c.689A>C was likely pathogenic (PM2_supporting, PM1, PM3_strong, PP4, BP4). We suggest that compound heterozygote variants p.Arg368Profs*22 and p.Gln230Pro in *GTPBP3* are causal for COXPD23 with recessive inheritance.

The patient also carried m.8108 A>G (100%) and m.10398A>G (100%) in mitochondria inherited from his mother (100%). The m.8108 A>G variant in MT-CO2 (Cytochrome c oxidase subunit II) occurred in a patient with Leigh syndrome who also carried three other variants (tRNA-Gln: m.4395A>G, ND6:m.14502T>C) [9]. The m.14502T>C variant was a common genotype of Leigh syndrome in the Chinese population. Therefore, the m.8108 A>G variant was independently related or unrelated to Leigh syndrome. In addition, evidence of the allele frequency of m.8108 A>G (0.12%, Mitomap, and 0.06%, gnomAD), APOGEE score (0.44 < 0.5, possibly benign), and pathogenicity predictions (Disease score 0.12 < 0.43, likely polymorphic) (https://www.hmtvar.uniba.it/query) indicated that it was neutral. Another variant, m.10398A>G of ND3 in our patient, was polymorphic due to the very high allele frequency found in high haplogroups at 50% or higher in Mitomap (http://www.mitomap.org/).

### 2.3. Systematic Analysis on Clinical Phenotypes of Individuals with GTPBP3 Variants

We collected the clinical assessment or medical records of 18 patients with *GTPBP3* variants in all clinical samples to analyze the relationships between genotypes and phenotypes (Appendix A). The average onset age was 1.7 years. Hyperlactatemia (100%) was present in all cases, 72% had cardiomyopathy, and 61% had abnormal signals on the brain MRI, which roughly corresponded to data reported in the previous literature. In addition to the three typical characteristics, COXPD23 patients may also present with other clinical phenotypes, such as visual impairment, seizures, global developmental delay systemic hypotonia, hyporeactivity, congestive heart failure, arrhythmia, feeding difficulties, etc.

According to Yan’s clinical classification [4], among the 18 patients, 7 were severe types and 11 were mild types. There were statically significant differences in the frequencies of developmental delay (*p* = 0.0023 **), arrhythmia (*p* = 0.0114 *), and congestive heart failure (CHF, *p* = 0.0491 *) between the two groups. Arrhythmia and CHF were significantly presented in the severe type, and CHF was a common cause of infant death among individuals with the severe type (Figure 3A). Moreover, we also found that the average age of onset of patients with homozygous variants (0.3 years old) was earlier than that of patients with compound heterozygotes (2.7 years old). Based on the comparison of the proportion of variant types between the above two groups, there was a significant difference (*p* = 0.0491 *) when comparing severe types with homozygous variants (5/7, 71.43%) with mild types (2/11, 18.18%) (Figure 3B). Briefly, we observed more homozygous variants in severe COXPD23. However, due to the small sample size of patients, further research is still required to support this view.

### 2.4. Variant Spectrum and Distribution of GTPBP3 in the Severe and Mild Type

In this study, we systematically analyzed the rare variants in gnomAD and COXPD23 patients (ClinVar and the related literature) (Table 1), divided into LOFVs (loss-of-function variants) and missense variants. As described in Figure 4, the proportion of LOFVs in *GTPBP3* from clinical samples (16/35, 45.7%) was significantly higher than that from the population group derived from gnomAD (20/225, 8.9%) (Figure 5A, *p* < 0.0001 ****).

As COXPD23 is an autosomal recessive disease, carriers with pathogenic variants are normal. Therefore, we observed there was no significant difference in the distribution of *GTPBP3* variants (LOFVs or Missense) across the five domains between COXPD23 patients and general populations (Figure 2B). When we only analyzed the homozygous missense variant in the coding region of *GTPBP3* from gnomAD and patients, we noticed significantly different distributions of the variants among patients located in the N-terminal region (*p* < 0.0001 ****) and the Trme-type G domain (*p* < 0.0001 ****) (Appendix A). The small sample size could lead to insufficient statistical analyses, and an expanded sample size is still required to support this view.

This study also examined the distribution of variants across domains between severe and mild types (Appendix A). We found that the variants from severe patients tended to be enriched in the Trme-type G domain for both LOFVs and missenses (Figure 5B), indicating that the G domain is crucial for the *GTPBP3* protein. In addition, the LOFVs in *GTPBP3* were frequently located in the N-terminal domain (especially in the transport peptide region), which might have disrupted most of the *GTPBP3* and lead to NMD (nonsense-mediated decay) for the severe type. At the same time, missense variants in *GTPBP3* frequently occurred in the central helical domain and G domain for severe patients (Figure 5B,C).

## 3. Discussion

We report a case of Combined Oxidative Phosphorylation Deficiency 23 caused by compound heterozygous variants in the *GTPBP3* gene. He carries two variants, a novel frameshift variant c.1102dupC (p.Arg368Profs*22) inherited from the maternal allele and a missense variant c.689A>C (p.Gln230Pro) inherited from the paternal allele. The patient developed clinical symptoms, such as hyperlactatemia, hypertrophic cardiomyopathy, self-limited epileptic seizures, feeding difficulties, intellectual disability and motor developmental delay, and abnormal visual development, and was diagnosed with COXPD23.

There are numerous studies that prove that the *GTPBP3* gene is associated with the post-transcriptional modification of mt-tRNA, and its deficiency affects the cellular oxygen consumption rate, ATP level, mitochondrial function, etc. [6,7,13]. Martinez-Zamora et al. (2015) reported that stable silencing of *GTPBP3* triggered the AMPK-dependent retrograde signaling pathway, resulting in reduced pyruvate transport, the promotion of a transition from pyruvate to fatty acid oxidation, and uncoupling of glycolysis and oxidative phosphorylation [14]. It results in lactic acidosis, the most common clinical feature in COXPD23 patients [15]. Simultaneously, these metabolic changes and the low ATP level may negatively affect the development and functioning of high-energy organs, such as the heart and brain. Afterward, Chen et al. (2019) established a zebrafish model with the knockout of *GTPBP3* and observed cardiac defects in both juvenile and adult zebrafish models that reproduced the clinical phenotype of HCM patients with *GTPBP3* variants [13]. However, the specific pathogenic mechanism of mitochondrial dysfunction caused by *GTPBP3* defect leading to hypertrophic cardiomyopathy remains unclear.

Five studies typically focused on COXPD23 with *GTPBP3* variants. Kopajtich et al. described 11 COXPD23 cases carrying 13 different variants of *GTPBP3* presented with respiratory chain complex deficiency, cardiomyopathy, lactic acidosis, and encephalopathy in 2014 [3]. Elmas et al. reported a 10-year-old girl with a *GTPBP3* variant who developed psychomotor developmental delay, seizures, hearing impairment, and delayed myelination in early childhood in 2019 [12]. Three cases of COXPD23 were first described in China by Yan et al. in 2021. They classified the previously reported cases into severe and mild types according to the different patients’ phenotypes [4]. In the same year, Zhao et al. and Yang et al. reported two mild Chinese cases carrying *GTPBP3* variants with similar clinical symptoms of mitochondrial disease, which provided new data for clinical diagnosis and genetic counseling for COXPD23 [10,11]. Referring to the classification of COXPD23 by Yan et al., we clarified the criteria of two types. The specific condition of the severe type was infant-onset illness and early death (<1 year), while patients with the mild type could survive in early childhood. According to the classification, we summarized the clinical features of 18 patients reported in the previous literature, including our case. We found that hyperlactatemia and lactic acidosis, cardiomyopathy, and encephalopathy were the typical clinical symptoms of COXPD23. Congestive heart failure and arrhythmia were often observed in the severe type, and the frequencies of developmental delay were significantly high in the mild type. Only mild patients developed visual impairment, self-limited epileptic seizures, and developmental delay (Figure 2A), which may be not seen in severe COXPD23 patients due to its rapid onset and premature death.

Studying the genotype and phenotype correlation, we noticed that the variant type and allelic status were significantly related to different kinds of COXPD23. We showed that the severe type likely carried homozygous variants (severe versus mild, 71.43% versus 18.18%), while the mild type tended to have compound heterozygotes in Figure 3B (Appendix A). Typically, LOFVs in the N-terminal (transport peptide region) only occurred in severe patients, suggesting that it was critical or the consequence of it resulted in the dysfunction of complete *GTPBP3*. Additionally, the LOFVs carried by the severe patients were distinctly located within the TrmE-type G domain. The LOFVs in central helical domain 1 could be related to either severe or mild COXPD23, but those in central helical domain 2 might only present mild clinical episodes (Figure 5B). For missense variants, we observed that the variants in the severe patients mainly occurred in the N-terminal region, central helical domain 1, and TrmE-type G domain, and much more in the last domain compared with the mild patient (Figure 5C).

Four variants were identified in severe patients with homozygous variants, including c.32_33delinsGTG, c.424G>A, c.665-2delA, and c.1009G>C, resulting in the deletion of the G domain, or a missense variant in the G domain or in the N-terminus of the *GTPBP3* protein. A total of two variants were identified in mild patients with homozygous variants, including c.770C>A and c.836C>T, which were missense variants of the G domain. Although we did not observe significant differences in the distribution of these variants, it is worth noting that the variants from severe patients tended to be enriched in the G domain, and missense variants or deletions within this domain were seen in 13 cases (13/18) (Appendix A). Therefore, we speculated that variants in the G domain were closely related to the functional impairment of the *GTPBP3* protein. The TrmE-type G domain retained a typical Ras-like protein fold. It contained the G1 to G4 motifs unique to all GTPases, which were involved in binding guanine nucleotides and Mg^2+^, hydrolyzing GTP, and regulating their functional status by controlling their own conformational changes [6].

Interestingly, the mild cases previously reported in China all carried the variant c.689A>C, including the proband reported here (Table 2). According to PGG.Han [8,16], the allele frequency of this variant in Han Chinese populations is 0.0002744, higher than that of 0.0001089 in East Asian people (only 1 in 9181 individuals), but none in other ethnic groups; in clinical samples from the Chinese population, it is up to 0.1389 (5/36 individuals), which indicates that the variants might have a high frequency in the Chinese Han population. Comparing the phenotype who carried the allele of c.689A>C, five patients had some consistent clinical symptoms: the onset of age was above one year, except the case in this study, which was during the neonatal period, and all of them were alive. Hypertrophic cardiomyopathy (LVH), hyperlactatemia, developmental delay, including movement impairment and language delay, and abnormal bilateral thalamus signals were the common symptoms. As mitochondria genome detection was performed only in our case, we could not have removed the effects of m.8108 A>G and m.10398A>G that the proband carried. However, they are possibly benign in Leigh syndrome due to their high population frequency.

In summary, here, we report a mild case with *GTPBP3* variants, identify a novel variant c.1102dupC (p.Arg368Profs*22), and explore the genetic pathogenesis and genotype–phenotype association of *GTPBP3* variants leading to COXPD23. This study provides new information for the clinical diagnosis and genetic counseling of patients with this disease. Indeed, there are still very few cases of COXPD23 that have been reported worldwide, and the statistical analysis of the minor patient size was limited, so further studies are needed to elucidate the specific pathogenetic mechanism.

## 4. Materials and Methods

### 4.1. Patient and Clinical Evaluation

The clinical evaluation and medical record of the patient in this subject were from Suzhou Municipal Hospital. The clinical assessment included a physical examination, routine laboratory investigations, biochemical analysis, echocardiography, and brain MRI. The brain MRI showed bilateral T2 hyperintensities in the thalamus, and heart ultrasound detected dilated cardiomyopathy and decreased contractility. Blood analysis showed profound metabolic acidosis, such as hyperlactatemia and hyperalaninemia, and suspected mitochondrial diseases. This study obtained the informed consent of the proband and his family members, and approval by the Ethics Committee of Suzhou Municipal Hospital.

### 4.2. Trio Whole-Exome Sequencing and Sanger Sequencing

Trio whole-exome sequencing was performed for the proband and his family on xGen Exome Hyb Panel 2.0 (51M) (Integrated DNA Technologies, Suzhou, China.) and whole-genome sequencing was performed only for the proband at Chigen using MGI DNBSEQ-T7 sequencing technology following the manufacturer’s protocol. The identification of variants (SNP and Indel) was conducted using the Genome Analysis Toolkit (GATK) (broad institute) and allele frequency (MAF < 0.01) filtered by gnomAD, the 1000 Genomes Project, and variant annotation of a family-based vcf file that had been described previously [16]. The analysis pipeline also included the copy number variation and variants in mitochondrial gene sequencing analysis. The pathogenicity of candidate variants was interpreted according to ACMG pathogenicity guidelines. The variants in GTPBP3 in this family were confirmed with Sanger sequencing. 

### 4.3. Phenotypic Analysis of Individuals with GTPBP3 Variants

*GTPBP3* variants were reported to cause mitochondrial disease COXPD23 for the first time in 2014 [3]. We curated the studies through PubMed and CNKI from December 2014 to December 2022 by searching the following key terms: *GTPBP3*, variants, and mitochondrial disease. We also included the bibliographic references of the retrieved studies and reviews. Finally, 17 COXPD23 patients with clinical characteristics carrying *GTPBP3* pathogenic variants were included in this study for genotype–phenotype correlation analysis. All patients’ clinical and genetic characteristics are summarized and analyzed in Appendix A. Despite the clinical heterogeneity of COXPD23, we classified it into two groups: severe type and mild type, referring to a previous study (Yan). The severe type of COXPD23 was infant onset and had inferior outcomes, such as early death before reaching 1 year old. The mild type usually presented in early childhood and patients may survive with severe hyperlactatemia.

### 4.4. Comprehensive Analysis of GTPBP3 Variants

In addition to the novel *GTPBP3* variants reported in this study, 42 pathogenic variants from ClinVar and clinical cases in the literature were included in the COXPD23 patient group. Seven hundred and ninety-one variants were collected from population-based databases, such as 1KGP, ExAC, and gnomAD. Following the data processing flowchart of An Y et al. [17], this study filtered low-quality variants, synonymous variants, or variants with MAF ≥ 0.001 and compiled a comprehensive list (Figure 6). Finally, we collected 19 missense variants and 16 LOFVs (loss of function variants) in the COXPD23 patient group, and 205 missense variants and 20 LOFVs in the gnomAD group.

### 4.5. Distribution of Variants in GTPBP3 Domains

The *GTPBP3* protein with 492 amino acids contains five domains: the transport peptide region (1–81), the N-terminal domain (35–152aa), the central helix domain formed by the intermediate region (155–251aa) and the C-terminal region (376–489aa), and the TrmE-type GTP binding (G) domain (249–416aa). A small region in front of the N-terminal domain is the mitochondrial targeting signal. According to the CDD database (website), the G domain of the *GTPBP3* protein contains ten conserved motifs, including 3 GTP/Mg^2+^ binding sites (1: 259, 261–264aa; 2: 353–354, 356aa; and 3: 376–378aa), switch I motif (271–287aa), switch II motif (302–312 and 314–328aa), and G1–G5 motifs (G1: 256–263aa; G2: 284aa; G3: 303–306aa; G4: 353–356aa; and G5: 376–378aa) (Figure 2B). This study examined the distribution of variants in different domains of *GTPBP3* in COXPD23 patients and gnomAD. As the effect of splicing variants on the protein domain is unknown, it was not included in the present study.

### 4.6. Statistical Analyses

Statistical analysis was performed by two-sided Fisher’s exact test for qualitative variables via GraphPad Prime 6.0. We used the two-sided p value for Fisher’s exact test calculated by two categories: with variants or without variants.

## Figures and Tables

**Figure 1 genes-14-00552-f001:**
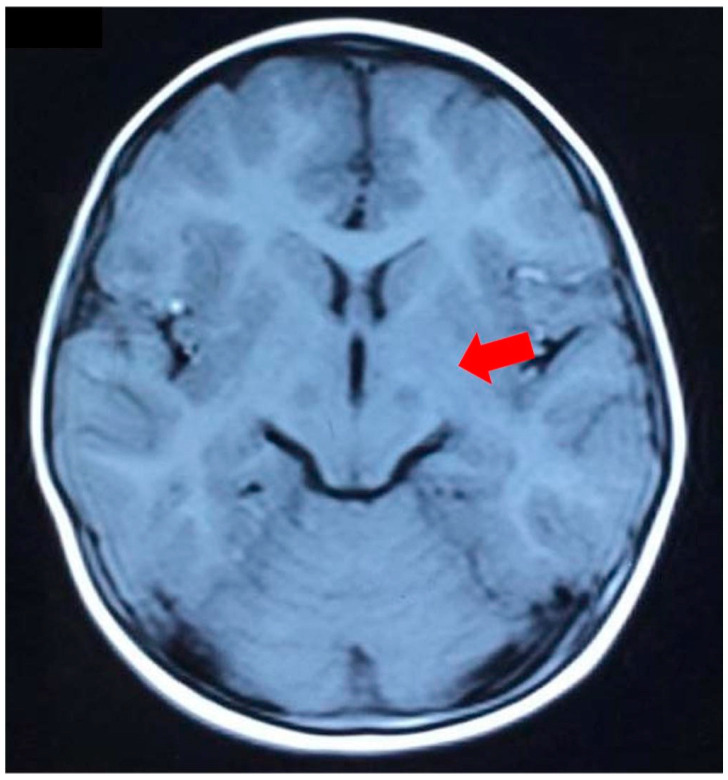
Brain MRI of our patient. MRI showed an abnormal signal shadow in the bilateral thalamus (red arrow).

**Figure 2 genes-14-00552-f002:**
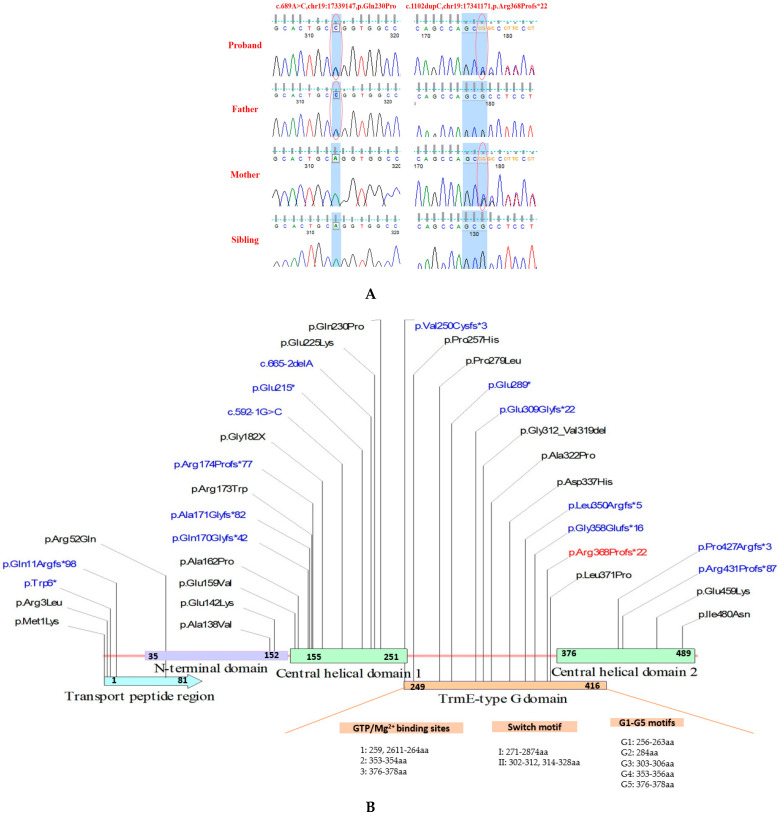
The pedigree and identified *GTPBP3* variants. (**A**) The variant c.689A>C (p.Gln230Pro) was detected in the proband and his father, and variant c.1102dupC (p.Arg368Profs*22) was detected in the proband and his mother. Both variants were shown in the red circle. (**B**) Distribution of 35 *GTPBP3* variants in clinical samples across the *GTPBP3* protein domains. The *GTPBP3* protein domains include the transport peptide region (light blue region), the N-terminal domain (light purple region), central helical domain 1 (light green region), the TrmE-type GTP-binding domain (orange region), and central helical domain 2 (C-terminal domain, light green region). The novel variants identified in this study are highlighted in red, loss-of-function variants in blue, and missense variants in black.

**Figure 3 genes-14-00552-f003:**
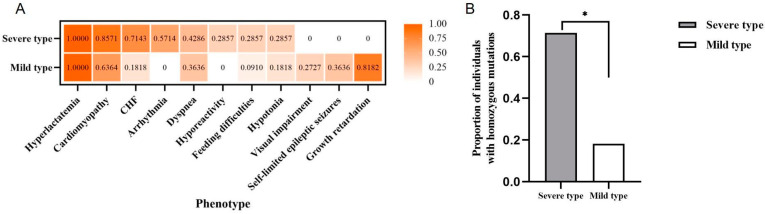
Comparison of phenotypes and variant types between severe and mild patients. (**A**) Comparing the frequencies of different phenotypes within two types of patients (*n* = 18); it was found that there were significant differences in the frequencies of developmental delay (*p* = 0.0023 **), arrhythmia (*p* = 0.0114 *) and CHF (*p* = 0.0491 *); (**B**) Comparing the proportion of individuals carrying homozygous variants in the two types of patients (*n* = 18), the proportion in severe patients (5/7, 71.43%) was significantly different from that in mild patients (20/225, 18.18%) (*p* = 0.0491 *). CHF: congestive heart failure. * < 0.05.

**Figure 4 genes-14-00552-f004:**
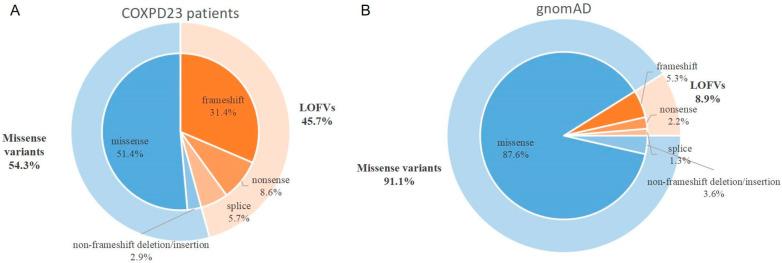
Proportions of different variant types in two groups. (**A**) COXPD23 patients (*n* = 35) included 11 frameshift variants, 3 nonsense variants, 2 splice variants, 1 non-frameshift deletion, and 18 missense variants; (**B**) The general population in gnomAD (*n* = 225) included 12 frameshift variants, 5 nonsense variants, 3 splice variants, 8 non-frameshift deletions/insertions, and 197 missense variants.

**Figure 5 genes-14-00552-f005:**
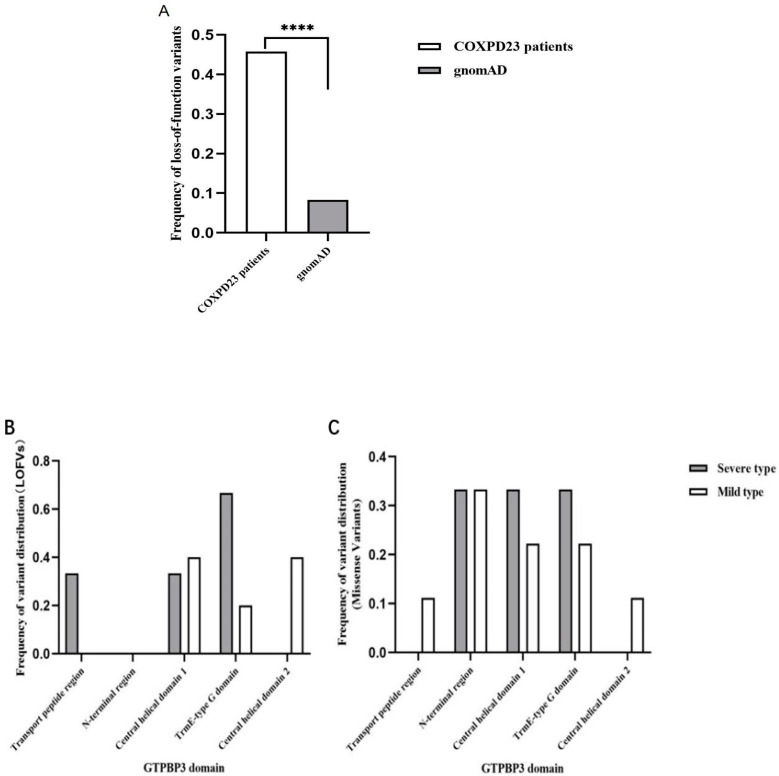
Comparison of the types of *GTPBP3* variants and their distributions in different domains. (**A**) Comparison of the frequency of loss-of-function variants between COXPD23 patients and general populations (*n* = 260) (*p* < 0.0001 ****); (**B**) Frequency of LOFV distribution in different domains between two groups (*n* = 8); (**C**) Frequency of missense variant distribution in different domains between two groups (*n* = 15).

**Figure 6 genes-14-00552-f006:**
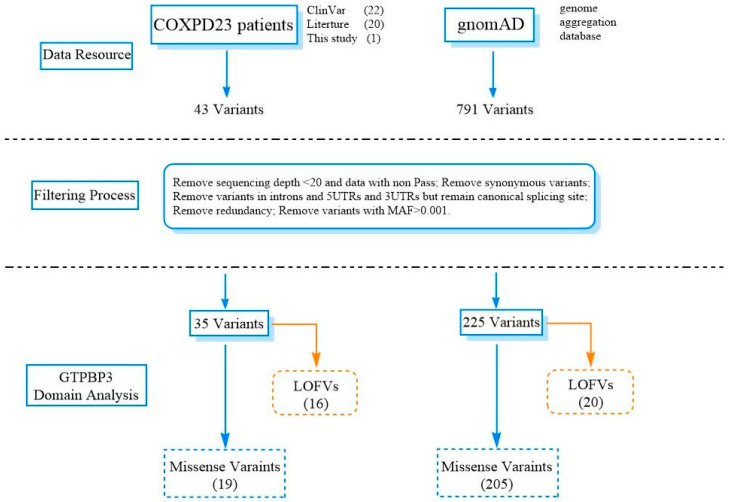
Workflow of the analysis of *GTPBP3* variants. Comprehensive list comprising 791 variants from gnomAD and 43 pathogenic variants from ClinVar and clinical cases in the literature. Finally, it includes 225 rare variants from gnomAD and 35 pathogenic variants from clinical cases. LOFVs, loss of function variants, include nonsense variants, frameshift deletion/insertion, and splicing variants. Missense variants included missense variants and non-frameshift deletion/insertion with unknown significance.

**Table 1 genes-14-00552-t001:** Comprehensive analysis of 35 variants in *GTPBP3.*

gDNA (GRCh38)	Ex.	Variant (NM_032620.4)	Mut. Type	Inheritance	Allele Freq.	SIFT	Polyphen-2	CADD	REVEL	Variants (In Trans)	AA Change (In Trans)	Related-Domain	Resources
chr19-17337613	Ex.1	c.2T>A, p.Met1Lys	Missense	UNK	n.d	D	D	D	0.342			T	ClinVar:488854
chr19-17337619	Ex.1	c.8G>T, p.Arg3Leu	Missense	UNK	0.0006418	D	D	D	0.096	c.934_957del	p.Gly312_Val319del	T	Kopajtich et al. (2014) [3]
chr19-17337628	Ex.1	c.17G>A, p.Trp6Ter	Nonsense	UNK	0.00001314							T/N/C1/G/C2	ClinVar:1405745
chr19-17337643-17337644	Ex.1	c.32_33delinsGTG, p.Gln11Argfs*98	Frameshift	UNK	n.d					Homo	Homo	T/N/C1/G/C2	Kopajtich et al. (2014) [3]
chr19-17338109	Ex.2	c.155G>A, p.Arg52Gln	Missense	UNK	0.00001604	D	D	D	0.635			N	ClinVar:372869
chr19-17338563	Ex.4	c.413C>T, p.Ala138Val	Missense	Paternal	0.00001196	D	D	D	0.580	c.509_510del	p.Glu170Glyfs*42	N	Yan et al. (2021) [4]
chr19-17338574	Ex.4	c.424G>A, p.Glu142Lys	Missense	Maternal	n.d	D	D	D	0.819	Homo/ c.689A>C	Homo/ p.Gln230Pro	N	Kopajtich et al. (2014) [3]; Yan et al. (2021) [4]
chr19-17338626	Ex.4	c.476A>T, p.Glu159Val	Missense	UNK	n.d	D	D	D	0.893	c.964G>C	p.Ala322Pro	N	Kopajtich et al. (2014) [3]
chr19-17338634	Ex.4	c.484G>C, p.Ala162Pro	Missense	Maternal	n.d	D	D	D	0.371	c.673G>A and c.964G>C	p.Glu225Lys and p.Ala322Pro	C1	Kopajtich et al. (2014) [3]
chr19-17338659-17338660	Ex.4	c.509_510del, p.Glu170Glyfs*42	Frameshift	Maternal	0.000006569					c.413C>T	p.Ala138Val	C1/G/C2	Yan et al. (2021) [4]
chr19-17338662	Ex.4	c.512del, p.Ala171Glyfs*82	Frameshift	UNK	n.d							C1/G/C2	ClinVar:1389465
chr19-17338667	Ex.4	c.517C>T, p.Arg173Trp	Missense	Paternal	0.000004009	D	D	D	0.300	c.643G>T	p.Glu215Ter	C1	ClinVar:488526
chr19-17338671-17338677	Ex.4	c.521_527del, p.Arg174Profs*77	Frameshift	UNK	n.d				0.256			C1/G/C2	ClinVar:1075628
chr19-17338694	Ex.4	c.544G>T, p.Gly182X	Missense	Paternal	n.d	-	-	D	-	c.689A>C	p.Gln230Pro	C1	Yan et al. (2021) [4]
chr19-17339005	Ex.5	c.643G>T, p.Glu215Ter	Nonsense	Maternal	n.d					c.517C>T	p.Arg173Trp	C1/G/C2	ClinVar:488527
chr19-17339131	Ex.6	c.673G>A, p.Glu225Lys	Missense	Paternal	0.00001601	T	T	T	0.018	c.484G>C	p.Ala162Pro	C1	Kopajtich et al. (2014) [3]
chr19-17339147	Ex.6	c.689A>C, p.Gln230Pro	Missense	Maternal	0.0001089	T	D	T	0.075	c.424G>A/c.544G>T /c.1073delG/c.1102dupC /c.1280delC	p.Glu142Lys/p.Gly182X /p.Gly358Glufs*16/p.Arg368Profs*22 /p.Pro427Argfs*3	C1	Yan et al. (2021) [4]; Yang et al. (2021) [10]; Zhao et al. (2021) [11]; Our patient
chr19-17339206	Ex.6	c.748del, p.Val250Cysfs*3	Frameshift	UNK	n.d							G/C2	ClinVar:1324519
chr19-17339228	Ex.6	c.770C>A, p.Pro257His	Missense	UNK	n.d	T	D	D	0.715	Homo	Homo	G	Kopajtich et al. (2014) [3]
chr19-17339461	Ex.7	c.836C>T, p.Pro279Leu	Missense	UNK	0.00008873	D	D	D	0.323	Homo	Homo	G	Eimas et al. (2019) [12]
chr19-17339490	Ex.7	c.865G>T, p.Glu289Ter	Nonsense	UNK	n.d							G/C2	ClinVar:642459
chr19-17339550	Ex.7	c.925dup, p.Glu309Glyfs*22	Frameshift	UNK	n.d							G/C2	ClinVar:1452744
chr19-17339559-17339582	Ex.7	c.934_957del, p.Gly312_Val319del	Non-frameshift deletion	UNK	n.d	-	-	-	-	c.8G>T	p.Arg3Leu	G	Kopajtich et al. (2014) [3]
chr19-17339589	Ex.7	c.964G>C, p.Ala322Pro	Missense	Paternal	0.0001104(total)	D	D	D	0.635	c.484G>C	p.Ala162Pro	G	Kopajtich et al. (2014) [3]
chr19-17341078	Ex.8	c.1009G>C, p.Asp337His	Missense	UNK	n.d	D	D	D	0.796	Homo	Homo	G	Kopajtich et al. (2014) [3]
chr19-17341118	Ex.8	c.1049del, p.Leu350Argfs*5	Frameshift	UNK	n.d							G/C2	ClinVar:280249
chr19-17341142	Ex.8	c.1073del, p.Gly358Glufs*16	Frameshift	Paternal	n.d					c.689A>C	p.Gln230Pro	G/C2	Yang et al. (2021) [10]
chr19-17341171	Ex.8	c.1102dup, p.Arg368Profs*22	Frameshift	Maternal	n.d					c.689A>C	p.Gln230Pro	G/C2	Our Patient
chr19-17341181	Ex.8	c.1112T>C, p.Leu371Pro	Missense	Paternal	0.000004009	T	D	D	0.314	c.440C>T	p.Ala147Val	G	ClinVar:488528
chr19-17341504	Ex.9	c.1280del, p.Pro427Argfs*3	Frameshift	Maternal	n.d					c.689A>C	p.Gln230Pro	C2	Zhao et.al. (2021) [11]
chr19-17341515	Ex.9	c.1291dup, p.Pro430Argfs*86	Frameshift	UNK	n.d							C2	ClinVar:180614
chr19-17341599	Ex.9	c.1375G>A, p.Glu459Lys	Missense	UNK	n.d	D	D	D	0.739	c.1291dupC	p.Pro430Argfs*86	C2	Kopajtich et al. (2014) [3]
chr19-17341663	Ex.9	c.1439T>A, p.Ile480Asn	Missense	UNK	n.d	D	D	D	0.515			C2	ClinVar:800909

D: Damage, T: Tolerated; UNK: unknown; n.d: did not observe; Related domain: T: transport peptide region; N: N-terminal region; C1: Central helical domain 1; C2: Central helical domain 2; G: TrmE-type G domain.

**Table 2 genes-14-00552-t002:** Comparison of clinical and genetic characteristics of patients carrying *GTPBP3* variant c.689A>C.

Patient No.	SP202010	#Case	#Case	#2 ^a^	#3 ^b^
	Our Patient	Zhao XX et al. (2021) [11]	Yang Q et al. (2021) [10]	Yan HM et al. (2021) [4]	Yan HM et al. (2021) [4]
Variants	c.1102dupC, p.Arg368Profs*22(mat.) c.689A>C, p.Gln230Pro(pat.) m.8108A>G (100%)	c.1280delC, p.Pro427Argfs*3(mat.) c.689A>C, p.Gln230Pro(pat.)	c.689A>C, p.Gln230Pro (mat.) c.1073delG, p.Gly358Glufs*16(pat.)	c.689A>C, p.Gln230Pro(mat.) c.544G>T, p.Gly182*(pat.)	c.424G>A, p.Glu142Lys(mat.) c.689A>C, p.Gln230Pro(pat.)
Gender	M	F	F	F	F
AO	3 days	9 years	3 years	1 year	1 year
Movement	Could walk but not run or jump	Could walk but had weakness in both lower limbs	Could walk slowly	Could not walk	Could walk or run but easily fell down
Language	Speak 5–6 words and unclear pronunciation	UNK	Speak a few words but no complete sentences	Could not speak	Speak a few words but no complete sentences
Other phenotypes	Intellectual disability and motor developmental delay; self-limited epileptic seizures; abnormal visual development; short stature	Heart failure; intellectual developmental delay; abdominal pain; abdominal distention	Respiratory failure; myocardial damage; stroke-like syndrome, global developmental delay	Developmental delay; hypotonia	Developmental delay; intellectual disability; fatigability
TTE	HCM (LVH)	HCM (LVH)	HCM (LVH)	UNK	HCM (LVH)
Brain MRI	Abnormal signal of bilateral thalamus	UNK	Abnormal signal of the bilateral cortical spinal tract	bilateral lesions in the midbrain, thalamus, and dentate body of the cerebellum	Bilateral lesions in the brain stem, thalamus, and dentate body of the cerebellum
Plasma	Lactate: 3–11.6 mmol/L↑ Normal amino acid and acylcarnitine profiles	Lactate: 5.53 mmol/L↑ N-terminal pro-BNP: 1603.3 pg/mL↑	Lactate: 29 mmol/L↑ CK-MB: 15.92μg/L, Pyruvate 599μml/L β-hydroxybutyrate Acid 0. 71 mmol/L, blood ammonia: 45.1μmol/L	Lactate: 7.7∼14 mmol/L↑ CK-MB: 32 U/L, Hyperalanine: 256.75 mmol/L normal acylcarnitine profiles	Lactate: 4.26–16 mmol/L↑
MS (Urine)	Increased levels of lactic acid, 3-hydroxybutyric acid, 3-methylglutaconic acid, 3-hydroxyglutaric acid, and hippuric acid	UNK	UNK	Increased level of Lactic acid,	Normal organic acid profile
EEG/ECG	High-amplitude 2–4 Hz sharp slow wave paroxysms mainly in the right occipital region, sometimes spreading to both frontal and occipital regions	UNK	Bilateral occipital rhythm dominated by 5–7 Hz θ rhythm, no paroxysmal EEG	UNK	Normal
Outcomes; cause of death	Alive after 10 years	Alive after 17 years	Alive after 3 years	Alive after 3 years	Alive after 3 years

Abbreviations are as follows: AO, age of onset; TTE, transthoracic echocardiography; LVH: left ventricular hypertrophy; MRI, magnetic resonance imaging; UNK, unknown; CHF, congestive heart failure; DCM, dilated cardiomyopathy; HCM, hypertrophic cardiomyopathy; LVH/RVH, left/right ventricular hypertrophy; WPW, Wolff–Parkinson–White syndrome; FTT, failure to thrive: MS: mass spectrum; Mat.: maternal; Pat.: paternal; * They have consanguineous parents. ^a,b^ These individuals are siblings. NCBI reference sequence: NM_032620.4.

## Data Availability

Data supporting reported results can be provided on request.

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
