# Peer review of "Pathogenicity Analysis of a Novel Variant in GTPBP3 Causing Mitochondrial Disease and Systematic Literature Review"

_genes, 2023, doi:10.3390/genes14030552_

Round 1

Reviewer 1 Report

This is an interesting article that expands the genotypic and phenotypic spectrum of COXPD23 and aims at drawing a first genotype-phenotype correlation. The case presented is interesting and relevant, and the bioinformatics is solid. There are, however, several points that should be corrected:

Major:

1.     “Studying the genotype and phenotype correlation, we noticed the mutation type and allelic status were significantly related to different kinds of COXPD23. We showed that the severe type likely carried homozygous mutations”. Although this was statistically significant, I do not believe that it truly reflects a biological fact, at least not without further experimentation (which is without the scope of this paper). The fact that variants are homozygous, by itself, reflects more the populational genetics rather than a biological mechanism. What really matters is the type of variant, and the specific variants, that are homozygous or not. In this way, I suggest mentioning it but not drawing biological conclusions from it, as it is most likely spurious. The location of the LOFVs, on the other hand, is very interesting. 

Minor:

1.     Gene names should follow the HUGO nomenclature guidelines. Please italicize all gene names. 

2.     “Retardation” is a pejorative word and should not be used. Please change as appropriately – for instance, with “intellectual disability” instead of “mental retardation”, “developmental delay” instead of “retardation of motor development”, “failure to gain appropriate weight” instead of “growth retardation”.

3.     Similarly, “mutation” can be offensive and should not be used. Please use only “variant” (or, as adequate, “pathogenic variant”). 

4.     There is a paucity of citations in the introduction (for instance, where did this prevalence data come from?). Please cite references appropriately.

5.     In the introduction, I would have liked to see a little bit more about the GTPBP3 itself – how it works, a little about its structure, its GTP-binding function. 

6.     Please remove all patient-identifying information such as birthday. 

7.     It is unclear how could the authors observe deficits in motor development and language at 3 days of age. Please clarify what was observed, as no milestones would be expected at this age. And please see point #2 above. 

8.     Please include more information on the patient’s seizures and which antiepileptic medication(s) was/were used. 

9.     Please include normal reference ranges for the lactate level. 

10.  Was an ECG performed? If so, please provide results and consider, if interesting, including the tracing as a figure.

11.  It would be interesting to see the chromatogram for the organic acids, if available. I suggest that the authors include it as a figure. Also, the interpretation as “nutritional disorders” is not standard and should be clarified.

12.  Please provide a reference for Yan’s clinical classification in loco (line 154), not only at the discussion.

13.  The last paragraph in the results section (lines 180-187) belongs in the discussion. 

14.  “Through the analysis of the distribution of the variants in the GTPBP3 structural domain and the ten conserved motifs, no significant difference between COXPD23 patients and general populations but more variants located in TrmE-type G domain in COXPD23 patients than in those from gnomAD.” This phrase is confusing and should be re-written. 

15.  In figure 1, I believe that the red arrow is pointing at the cerebellum, not at the thalamus as the legend claims. Please revise.

16.  In figure 2, the pedigree is not really adding much information. It may be removed. 

Author Response

  1. “Studying the genotype and phenotype correlation, we noticed the mutation type and allelic status were significantly related to different kinds of COXPD23. We showed that the severe type likely carried homozygous mutations”. Although this was statistically significant, I do not believe that it truly reflects a biological fact, at least not without further experimentation (which is without the scope of this paper). The fact that variants are homozygous, by itself, reflects more the populational genetics rather than a biological mechanism. What really matters is the typeof variant, and the specific variants, that are homozygous or not. In this way, I suggest mentioning it but not drawing biological conclusions from it, as it is most likely spurious. The location of the LOFVs, on the other hand, is very interesting. 

Response:  It is a very good point. We agree with the reviewer that the proportion of homozygosity of pathogenic variants in patients reflects the number of carriers who have the same variants. In this case, we observed the proportion of homozygous mutation (71.4%) typically in the severe type, which might further enhance Loss-of-function effect of GTPBP3 defect. For other genes with a genetically dominant negative effect, such as GRIN1 or SLC1A2, the patients with the biallelic variants were more severe than the ones with homozygous variants. Sometimes the homozygous was lethal and we could not find them alive, which might indicate the underlined mechanism we would not ignore. In this study, we only described the observation and could not draw a conclusion about its underlined biological mechanism. Of course, we also agree with the reviewer about it is important for the type of variants and location of LOFVs.

Minor:

  1. Gene names should follow the HUGO nomenclature guidelines. Please italicize all gene names. 

Response: Thank you for the reviewer’s reminder. We checked the HUGO nomenclature and GTPBP3 was the correct name. We also changed all gene names in italics.

  1. “Retardation” is a pejorative word and should not be used. Please change as appropriate – for instance, with “intellectual disability” instead of “mental retardation”, “developmental delay” instead of “retardation of motor development”, “failure to gain appropriate weight” instead of “growth retardation”.

Response: We thank the reviewer for pointing out this point. We have changed “Retardation” into proper words like “developmental delay”, “intellectual disability” etc.

  1. Similarly, “mutation” can be offensive and should not be used. Please use only “variant” (or, as adequate, “pathogenic variant”). 

Response: We thank the reviewer for pointing out this point. We have changed “mutations” into “variants”。

  1. There is a paucity of citations in the introduction (for instance, where did this prevalence data come from?). Please cite references appropriately.

Response: We thank the reviewer for pointing out this point. We have added the reference appropriately.

  1. In the introduction, I would have liked to see a little bit more about the GTPBP3 itself – how it works, a little about its structure, its GTP-binding function. 

Response: Thanks. We have added a description of the biological function and structure of GTPBP3.

  1. Please remove all patient-identifying information such as birthdays. 

Response: Thanks, we have removed the patient-identifying information.

  1. It is unclear how could the authors observe deficits in motor development and language at 3 days of age. Please clarify what was observed, as no milestones would be expected at this age. And please see point #2 above. 

Response: This is indeed a good question. The initial symptom was feeding difficulties on the third day after birth and was improved after hospitalization. Afterward, he was observed with motor, intellectual, and language developmental delays compared to his peers. After 16 months of life, the child could walk and run independently but unstably and could not bounce. We noticed no language development at the age of 3 years. We have revised the paragraph to clarify the milestones of the boy’s development.

  1. Please include more information on the patient’s seizures and which antiepileptic medication(s) was/were used. 

Response: He had seizures since eight months of birth and had taken anti-epileptic medication (Eitracetam) for six years. At six years old, the EEG revealed that the high-amplitude 2-4hz sharp waves and slow waves distributed primarily in the right occipital portion were sporadic, occasionally affecting the bilateral frontal and occipital parts (supplemental Figure S1). And no seizure episodes happened since the seven years old.

  1. Please include normal reference ranges for the lactate level. 

Response: Thanks, we have added the normal ranges of the lactate level (normal: 0-2.2mmol/l) into the manuscript.

  1. Was an ECG performed? If so, please provide results and consider, if interesting, including the tracing as a figure.

Response: We did not perform ECG (Electrocardiography). But echocardiography at 8.5 years showed a significant thickening of the left ventricular wall, with a thickness of 14.7mm compared with 5.5mm 5 years earlier. It is the specific clinical feature for individuals with GTPBP3 variants.

  1. It would be interesting to see the chromatogram for the organic acids, if available. I suggest that the authors include it as a figure. Also, the interpretation as “nutritional disorders” is not standard and should be clarified.

Response: Thanks. We have included the figure of the chromatogram for the organic acids as supplemental material (Figure S2). But it is very vigorous and could not be put into context. 

  1. Please provide a reference for Yan’s clinical classification in loco (line 154), not only at the discussion.

Response: Thanks. We have cited Yan’s clinical classification in the 4th of reference.

  1. The last paragraph in the results section (lines 180-187) belongs in the discussion. 

Response: In 2.4 part of our result, we utilized the curated variants of GTPBP3 to complete the two aspects of variant spectrum analysis: case vs control, the severe vs the mild. The last paragraph in the results section (lines 180-187) was about the comparison between the severe and mild types. That is the description of the analysis. In the discussion section, we also discuss the potential possibility or interpretation of the results in the third paragraph.

  1. “Through the analysis of the distribution of the variants in the GTPBP3 structural domain and the ten conserved motifs, no significant difference between COXPD23 patients and general populations but more variants located in TrmE-type G domain in COXPD23 patients than in those from gnomAD.” This phrase is confusing and should be re-written. 

Response: We agree with the reviewer. This sentence was revised: No significant difference was found between COXPD23 patients and general populations across the structural domain and the ten conserved motifs of GTPBP3.

  1. In figure 1, I believe that the red arrow is pointing at the cerebellum, not at the thalamus as the legend claims. Please revise.

 Response: Thank you for the reviewer’s suggestion. We have revised figure1.

  1. In figure 2, the pedigree is not really adding much information. It may be removed. 

Response: Thank you for the reviewer’s suggestion. We removed the pedigree.

Reviewer 2 Report

The article is devoted to an extremely rare autosomal recessive disease. Of course, the accumulation and generalization of clinical and molecular genetic information is extremely important for medical geneticists. Although the article contains little new information from a scientific point of view, a good review of already published cases can make it valuable for the reader.

I have comments on the presentation and processing of data.

1)It seems to me that within the framework of the small number of cases described, there is no need to compare homozygotes and compound heterozygotes. I propose to focus on the type of mutations, for example, two mutations with loss of function, one with loss of function and a missense, two missense mutations.

2) Why did the authors compare the spectrum of mutations in patients and in gnomad? All this information is contained in the Gene constraint section https://gnomad .broadinstitute.org/gene/ENSG00000130299?dataset=gnomad_r2_1 I propose to remove this comparison from the work. 3) It is necessary to reconsider the expediency of distributing Log options by domains. The severity of these options must be considered in relation to the possible NMD.

4) A more detailed description of the statistics used to compare such small groups is needed.

Author Response

It seems to me that within the framework of the small number of cases described, there is no need to compare homozygotes and compound heterozygotes. I propose to focus on the type of mutations, for example, two mutations with loss of function, one with loss of function and a missense, two missense mutations.

Response: Thank for reviewer’s suggestion. We also consider the analysis on the type of mutations, however, no any useful findings due to small sample size.

Why did the authors compare the spectrum of mutations in patients and in gnomad? All this information is contained in the Gene constraint section https://gnomad .broadinstitute.org/gene/ENSG00000130299?dataset=gnomad_r2_1 I propose to remove this comparison from the work.

Response: Thank for the reviewer’s suggestion. Compared to the gnomAD, We would like to know if the variants of GTPBP3 in clinical cases had any special properties such as location, variant type, and distribution in the function domain. We removed the analysis of variant distribution for LOFVs and missense variants between the two groups. But we preserved analysis of the proportion of LOFVs and missense variants in GTPBP3 between patients and gnomAD.

It is necessary to reconsider the expediency of distributing Log options by domains. The severity of these options must be considered in relation to the possible NMD.

Response: Thank for the reviewer’s suggestion. We agree with the reviewer. We want to identify which domain was not tolerated by disruption mostly caused by LOFVs. We observed the LOFVs in GTPBP3 frequently located in the N-terminal domain which might disrupt most of GTPBP3 and lead to NMD (nonsense mediate decay) for the severe type.

A more detailed description of the statistics used to compare such small groups is needed.

Response: Thank for the reviewer’s suggestion. For a small sample size, only Fisher’s exact test would be appreciated. Especially, one or more of the cell counts in a 2X2 table is less than 5. We used a two-sided p-value for Fisher’s exact test calculated by two categories: with the variants or without variants. We revised the part of the statistical analysis.

Reviewer 3 Report

The authors reported that a boy with COXPD23 carries compound heterozygous mutations in the GTPBP3 gene, and summarized features of previously reported CPOXPD23 patients. Since the disease is very rare, this case report with a summary of the disease will be a help of clinical doctors.

Minor points

1.     The authors showed a schematic protein structure of GTPBP3 in Fig.5. I recommend the authors to show the structure in Figure 2. Then the readers can easily understand positions of the mutations and the domains.

2.     I’m not sure how deeply the authors understand mechanism of nonsense mediate mRNA decay (NMD). NMD is often occurred when premature stop codons generated by mutations appear exons other than the last exon. Since the c.1102dupC mutation is in exon 8, not in the last exon (exon 9), the mutation likely causes NMD. I would like the authors to deepen their discussion regarding the nonsense mutations found in the GTPBP3 gene.

3.     If possible, the authors had better to examine mRNA levels and protein levels of GTPBP3 in the patient’s cells by RT-PCR and Western blotting to see whether the c.1102dupC mutation results in truncated protein or NMD.

4.     Page 3 line 142: I guess that “neural” is a typo of “neutral”.

5.     Pages 3, 5, and 7: “server type” is a typo of “severe type”.

Author Response

Minor points

  1. The authors showed a schematic protein structure of GTPBP3 in Fig.5. I recommend the authors to show the structure in Figure 2. Then the readers can easily understand positions of the mutations and the domains.

Response: Thank for the reviewer’s suggestion. We moved the schematic protein structure of GTPBP3 into Figure 2, which made the readers easily see the location of the variants.

  1. I’m not sure how deeply the authors understand mechanism of nonsense mediate mRNA decay (NMD). NMD is often occurred when premature stop codons generated by mutations appear exons other than the last exon. Since the c.1102dupC mutation is in exon 8, not in the last exon (exon 9), the mutation likely causes NMD. I would like the authors to deepen their discussion regarding the nonsense mutations found in the theGTPBP3

Response: Nonsense-mediated decay (NMD) is a cellular process of mRNA surveillance that removes aberrant and structurally defective transcripts before they are translated into proteins. The variant c.1102dupC (p. Arg368Profs*22) in exon 7 of GTPBP3 was predicted nonsense-mediated decay (NMD). We added this information in the manuscript.

  1. If possible, the authors had better to examine mRNA levels and protein levels of GTPBP3 in the patient’s cells by RT-PCR and Western blotting to see whether the c.1102dupC mutation results in truncated protein or NMD.

Response: Thank the reviewer’s suggestions. According to of ACMG guidelines on the interpretation of sequence variants, the variant c.1102dupC in GTPBP3 was pathogenic. The patient with the compound heterozygous variants in GTPBP3 had typical clinical features of GTPBP3-related monogenic disorder. In addition, the expression of GTPBP3 in the blood is very low (GTEx data). Now it is not available to get patients’ cells.   

  1. Page 3 line 142: I guess that “neural” is a typo of “neutral”.

Response: Thanks. It had been revised in the manuscript.

  1. Pages 3, 5, and 7: “server type” is a typo of “severe type”.

Response: Thanks. It had been revised in the manuscript.

Round 2

Reviewer 2 Report

The authors have made significant changes to the report. The work has improved significantly. There are few cases of COXPD23 registered in the world and the accumulation of information about genotypes and phenotypes of patients is very important for medical genetics. I would recommend a very careful editorial edit. There are a lot of typos in the text. Figure 6 was moved to Figure 1, while it remained in the manuscript and so on.

Author Response

The authors have made significant changes to the report. The work has improved significantly. There are few cases of COXPD23 registered in the world and the accumulation of information about genotypes and phenotypes of patients is very important for medical genetics. I would recommend a very careful editorial edit. There are a lot of typos in the text. Figure 6 was moved to Figure 1, while it remained in the manuscript and so on.

Response: Thank the reviewer for the suggestions.  We have checked typos and revised them carefully. Figure 6 was moved to Figure 1 while Figure 7 was moved to Figure 6. So you see Figure 6 remained in the manuscript but it is not the original figure 6.